# Towards understanding the antagonistic activity of phytic acid against common foodborne bacterial pathogens using a general linear model

Ines Boukhris[1], Slim Smaoui[1], Karim Ennouri[2], Nawres Morjene[1], Ameny Farhat-Khemakhem[1], Monia Blibech[1], Othman A. Alghamdi[3], Hichem Chouayekh ®[1,3]*

1 Laboratoire de Microorganismes et de Biomolécules, Centre de Biotechnologie de Sfax, Université de Sfax, Sfax, Tunisia, 2 Laboratory of Amelioration and Protection of Olive Genetic Resources, Olive Tree Institute, University of Sfax, Sfax, Tunisia, 3 Department of Biological Sciences, Faculty of Sciences, University of Jeddah, Jeddah, Kingdom of Saudi Arabia

* hschouayekh@uj.edu.sa

**Data Availability Statement:** All relevant data are within the manuscript and its Supporting Information files

## Abstract

The increasing challenge of antibiotic resistance requires not only the discovery of new antibiotics, but also the development of new alternative approaches. Therefore, in the present study, we investigated for the first time the antibacterial potential of phytic acid (*myo*-inositol hexakisphosphate, IP6), a natural molecule that is 'generally recognized as safe' (FDA classification), against the proliferation of common foodborne bacterial pathogens such as *Listeria monocytogenes*, *Staphylococcus aureus* and *Salmonella* Typhimurium. Interestingly, compared to citric acid, IP6 was found to exhibit significantly greater inhibitory activity ($P<0.05$) against these pathogenic bacteria. The minimum inhibitory concentration of IP6 varied from 0.488 to 0.97 mg/ml for the Gram-positive bacteria that were tested, and was 0.244 mg/ml for the Gram-negative bacteria. Linear and general models were used to further explore the antibacterial effects of IP6. The developed models were validated using experimental growth data for *L. monocytogenes*, *S. aureus* and *S.* Typhimurium. Overall, the models were able to accurately predict the growth of *L. monocytogenes*, *S. aureus*, and *S.* Typhimuriumin Polymyxin acriflavine lithium chloride ceftazidime aesculin mannitol (PAL-CAM), Chapman broth, and xylose lysine xeoxycholate (XLD) broth, respectively. Remarkably, the early logarithmic growth phase of *S.* Typhimurium showed a rapid and severe decrease in a period of less than one hour, illustrating the bactericidal effect of IP6. These results suggest that IP6 is an efficient antibacterial agent and can be used to control the proliferation of foodborne pathogens. It has promising potential for environmentally friendly applications in the food industry, such as for food preservation, food safety, and for prolonging shelf life.

**Funding:** The Tunisian Government provided support via the Contract Program LMB-CBS (2015-2018) which includes several research projects, including this work. Funding was also received from the recently accepted International Cooperation Grant Research Project N° UJ-02-009-ICGR, funded by the University of Jeddah, though the grant is not specific to this study, but rather focusing on bio-composites for active packaging. These provided funds were used to support the editing by a commercial and professional editing service as well as the publication fees of this manuscript in PLOS ONE.

**Competing interests:** The authors have declared that no competing interests exist

## 1. Introduction

Food safety is an important public health priority. Each year, around a third of the world population is infected by foodborne pathogens [1, 2]. The highest incidence of foodborne diseases occurs in Africa [1], where more than 91 million people are estimated to fall ill each year, leading to around 137 000 deaths [2]. *L. monocytogenes* is considered to be one of the most hazardous foodborne bacterial pathogens [3, 4]. It is particularly problematic, as it can form biofilms and survive for long periods of time in a food processing environment [5]. *L. monocytogenes* causes human listeriosis, a rare disease associated with high rates of hospitalization and mortality [6, 7]. The infection is mainly linked to the consumption of contaminated ready-to-eat foods, such as cheese and other dairy products, processed meats, salads, seafood, and raw eggs [8, 9]. Another bacterium, *S. aureus*, which belongs to the Gram-positive Micrococcaceae family, is considered to be one of the most common causes of foodborne disease in the world [10]. This bacterium causes gastrointestinal illness by secreting a range of toxins, including staphylococcal enterotoxins [10–12]. *S. aureus* is also able to form biofilms on food-contact surfaces, which highly increases its stress tolerance and, thus, its persistence in food-related environments [13–16]. *Salmonella* is also considered to be among the most common foodborne pathogens [17]. It causes Salmonellosis, a disease that is linked to the consumption of contaminated meats, especially poultry products [17]. This disease is responsible for the largest number of hospitalizations and deaths due to foodborne pathogens [18].

In order to prevent the transmission of foodborne disease, the food industry uses a variety of methods, which can be physical (temperature, radiation), chemical (bleach, alcohol, iodine) or chemotherapeutic (antibiotics) [19, 20]. However, several studies have suggested that synthetic sanitizers can have significant side effects, such as bleaching and the formation of toxic compounds [21]. In addition, the growing rate of antimicrobial resistance in foodborne bacterial pathogens is also becoming a major concern for food safety and one of the disquieting threats to global human health [22]. For this reason, the World Health Organization (WHO) recommended that breeders and the food industry stop misusing antibiotics, such as for promoting animal growth and preventing disease in healthy animals, in order to preserve the effectiveness of antibiotics for human medicine [22]. Moreover, the increasing demand for organic food has increased the interest in substituting these chemicals with natural products, which do not damage the host or the environment [23, 24]. Thus, studies of natural compounds with antimicrobial properties are warranted. There is currently much interest in the use of organic acids as 'environmentally friendly' sanitizers or preservatives [25], most notably citric acid, acetic acid and lactic acid [26]. Studies have mainly focused on citric acid, which can be used alone or alongside conventional sanitizers and other alternative technologies [27]. It is especially useful for fruit, such as strawberries [27]. However, it has been noted that organic acids, such as formic acid, lactic acid, propionic acid and their salts, often fail to provide the desired result when applied in practice, such as when used for decontaminating animal feed, thus occasioning substantial additional costs for operators (feed producers and farmers) [28]. Therefore, there is currently much interest in research on alternative organic acids that are both cheap and safe to use.

*Myo*-inositol 1, 2, 3, 4, 5, 6-hexakisphosphate (IP6), commonly known as phytic acid, is a naturally occurring compound that is 'generally recognized as safe', according to the U.S. Food and Drug Administration classification [29]. It represents the principal storage form of phosphorus (P) in whole cereals and other edible vegetable seeds, such as legumes and nuts, and may account for 65–85% of the total P in seeds [29]. IP6 is a negatively charged compound that has been considered an anti-nutrient since it acts as a strong chelator of vital minerals like calcium, iron, magnesium, copper, zinc, and potassium, reducing their absorption and bioavailability [30].

However, several recent studies in both humans and animals have demonstrated that this natural molecule acts as a disease-preventing compound [31]. Indeed, IP6 displays a broad range of pharmaceutical properties, including antioxidant [32, 33], neuroprotective [34, 35], anti-inflammatory [36], lipid lowering [37], pathological calcification preventing [38, 39] and anticancer activities [40–42]. In addition, Kim and Rhee [21] described the anti-biofilm effect of IP6 against *E. coli* O157:H7, especially when combined with NaCl. They suggested that a sanitizer that combines these two naturally occurring antimicrobial agents could be used by food safety managers who encounter thick biofilm formation in food processing environments.

The purpose of the present work is to assess the effectiveness of IP6 against several foodborne bacterial pathogens for the first time. This is achieved by: (i) determining the minimum inhibitory concentration (MIC) as compared to citric acid (CA); (ii) measuring the inhibition diameters of each indicator bacteria growth inhibition or reduction, and (iii) illustrating the mode of action of IP6 for inhibiting pathogen growth.

## 2. Materials and methods

### 2.1. Substrates and chemicals

Phytic acid solution [*myo*-Inositol hexakis (dihydrogen phosphate); IP6] was purchased from Sigma-Aldrich (593648; 50% (w/w) in water). Citric acid (CA), ($C_6H_8O_7$; CAS: 77-92-9), was purchased from Fluka, Switzerland.

### 2.2. Bacterial strains, media and culture conditions

The target bacterial strains were obtained from international culture collections (ATCC). They included Gram-positive bacteria: *Listeria monocytogenes* ATCC 19117 and *Staphylococcus aureus* ATCC 6538; and Gram-negative bacteria: *Salmonella* Typhimurium ATCC 14028, *Pseudomonas aeruginosa* ATCC 49189 and *Escherichia coli* ATCC 8739. These strains were used as indicator microorganisms for the antibacterial activity assays. *L. monocytogenes* ATCC 19117 was cultured on Polymyxin Acriflavin Lithium-Chloride Ceftazidime Aesculin Mannitol (PALCAM, LAB M Ltd, U.K) at 37°C for 24 h. *S. aureus* ATCC 6538 was grown on Chapman medium (Oxoid, Basingstoke, Hampshire, UK) at 37°C for 24 h, and *S.* Typhimurium ATCC 14028 was cultivated on Xylose Lysine Deoxycholate (XLD, Oxoid) at 37°C for 24 h. For the antagonist tests, the final inoculum concentration used for each indicator bacterium was $10^6$ colony-forming units of bacteria per milliliter (CFU/ml) as used in the method described by Smaoui et al. [24].

### 2.3. Determination of minimum inhibitory concentrations

The minimum inhibitory concentration (MIC) values, representing the lowest concentration of IP6 and CA at which the microorganism did not demonstrate visible growth after incubation, were determined against a panel of five bacteria, as described by Gulluce et al. [43] with minor modifications. The test was performed in sterile 96-well microplates with a final volume of 100 μl per well. A commercial stock solution of IP6 (50% (w/w) in water) and a stock solution of CA at 50% (w/w) in water were used. Then, the corresponding concentrations of IP6 and CA were transferred to each successive well in order to obtain a two-fold serial dilution of the original sample. In fact, each sample was dissolved to a final concentration of 0.078, 0.156, 0.312, 0.625, 1.25, 2.5, 5, 10 and 20 mg/mL and then filtered through 0.22 μm pore-size black polycarbonate filters (Millipore). To each test well 10 μl of cell suspension were added to final inoculum concentration of $10^6$ CFU/ml of bacterium. Positive growth control well consisted of *Listeria monocytogenes* ATCC 19117, *Staphylococcus aureus* ATCC 6538 and *Salmonella* Typhimurium ATCC 14028 respectively growth in PALCAM, Chapman and XLD. Plates were

then covered with the sterile plate covers and incubated at 37˚C for 24 h. As an indicator of microorganism growth, 25 μl of thiazolyl blue tetrazolium bromide (MTT) indicator solution (0.5 mg/ml) dissolved in sterile water was added to the wells and incubated at 37˚C for 30 min. The colourless tetrazolium salt acts as an electron acceptor and was reduced to a red-coloured formazan product by biologically active organisms. Where microbial growth was inhibited, the solution in the well remained clear after incubation with MTT. The determination of MIC values was done in triplicate.

## 2.4. Agar diffusion method

The antimicrobial activity of IP6 was evaluated by means of agar-well diffusion assays, as described by Valgas et al. [44]. Fifteen milliliters of the molten agar (45˚C) were poured into sterile petri dishes (Ø 90 mm). Working cell suspensions were prepared at $10^6$ CFU/mL, and 100 μl was evenly spreaded onto the surface of the agar plates of Luria–Bertani (LB) agar (Oxoid Ltd, UK). Once the plates had been aseptically dried, 06 mm wells were punched into the agar with a sterile Pasteur pipette. IP6 was dissolved in water to a final concentration of 50 mg/ml and then filtered through 0.22 μm pore-size black polycarbonate filters (Millipore). Thus, 50 μl were placed into the wells and the plates were incubated at 37˚C for 24 h for bacterial strains. Antibacterial activity was evaluated by measuring the diameter of circular inhibition zones around the well. The un-inoculated media were also tested for inhibitory zones as a control. Tests were performed in triplicate.

## 2.5. Mode of action of phytic acid

The bacteriostatic or bactericidal mode of action of IP6 was tested using a method described previously by Jiang et al. [45]with some modifications. *L. monocytogenes* ATCC 19117, *S. aureus* ATCC 6538 and *S.* Typhimurium ATCC 14028 were cultivated in 100 mL of PALCAM, Chapman broth and xylose lysine xeoxycholate (XLD) broth, respectively. After 3 h of incubation, the bacterial growth reached the beginning of the exponential phase (about $10^6$ CFU/ml). At this moment, IP6 was added to the cultures at a final concentration of 1×MIC, 2×MIC and 4×MIC. The three indicator strains grown in the absence of IP6 were used as controls. Changes in the turbidity of the cultures were recorded at $OD_{600 \text{ nm}}$ and the number of CFU/mL was determined by plating the serial decimal dilutions of samples on PALCAM agar for *L. monocytogenes* ATCC 19117, Champan agar for *S. aureus* ATCC 6538, and XLD agar for *S.* Typhimurium ATCC 14028, and then counting the colonies that appeared.

## 2.6. Statistical analyses

Measurements were carried out in triplicate and repeated three times. A one-way analysis of variance (ANOVA) was run for each parameter using the SPSS 19 statistical package (SPSS Ltd., Woking, UK). Means and standard errors were calculated and a probability level of $P < 0.05$ was used for assessing the statistical significance of the experimental data. Tukey's *post hoc* test was used to determine whether differences between each of the mean values were significant ($P < 0.05$). Plate count data were converted to logarithms prior to the statistical analyses. Linear mixed models were used, which made certain assumptions about the errors (e.g. constant variance), to compare the CFU values among treatments with different incubation times (measured in h). Mixed models were fitted using SPSS 19 and followed by *post hoc* contrasts through the origin. The interpretation of the statistical output by analysis of covariance (ANCOVA, SPSS; covariates, time and trial) of a mixed model requires an understanding of how to explain the relationships among the fixed and random effects in terms of the hierarchy levels. The significance or not of all estimates was confirmed by Wald Z.

## 3. Results and discussion

### 3.1. Assessment of the antibacterial activity of phytic acid

The antibacterial activity of phytic acid (IP6) was evaluated against Gram-positive (*L. monocytogenes* ATCC 19117 and *S. aureus* ATCC 6538) and Gram-negative (*S.* Typhimurium ATCC 14028, *P. aeruginosa* ATCC 49189 and *E. coli* ATCC 8739) foodborne bacterial pathogens. The antagonistic activity was assessed by determining the MIC values in comparison with CA and by measuring the inhibition zones by agar diffusion method.

**3.1.1 Determination of MIC values of phytic acid and citric acid.** The comparison of the antimicrobial activity of IP6 and CA against both Gram-negative and Gram-positive bacteria, as determined using the MIC values, is illustrated in **Fig 1**. We found that the IP6 MIC ranged from 0.244 to 0.976 mg/ml, while the CA MIC ranged from 1.25 to 2.5 mg/ml. For the Gram-positive bacteria, the IP6 MIC and the CA MIC ranged from 0.488 to 0.976 mg/ml and from 1.25 to 2.5 mg/ml, respectively. The lowest MIC values were for *S. aureus* ATCC 6538. For the Gram-negative bacteria, the IP6 MICs were reduced by at least 50%, while the CA MIC remained the same as that observed for *S. aureus* ATCC 6538 (1.25 mg/ml). No significant ($P > 0.05$) difference was observed in terms of the IP6 MIC and CA MIC (0.244 mg/ml and 1.25 mg/ml, respectively) between the three Gram-negative bacteria (**Fig 1**).

These results indicate that IP6 is more efficient than CA against foodborne pathogenic bacteria (Gram-positive and Gram-negative bacteria).

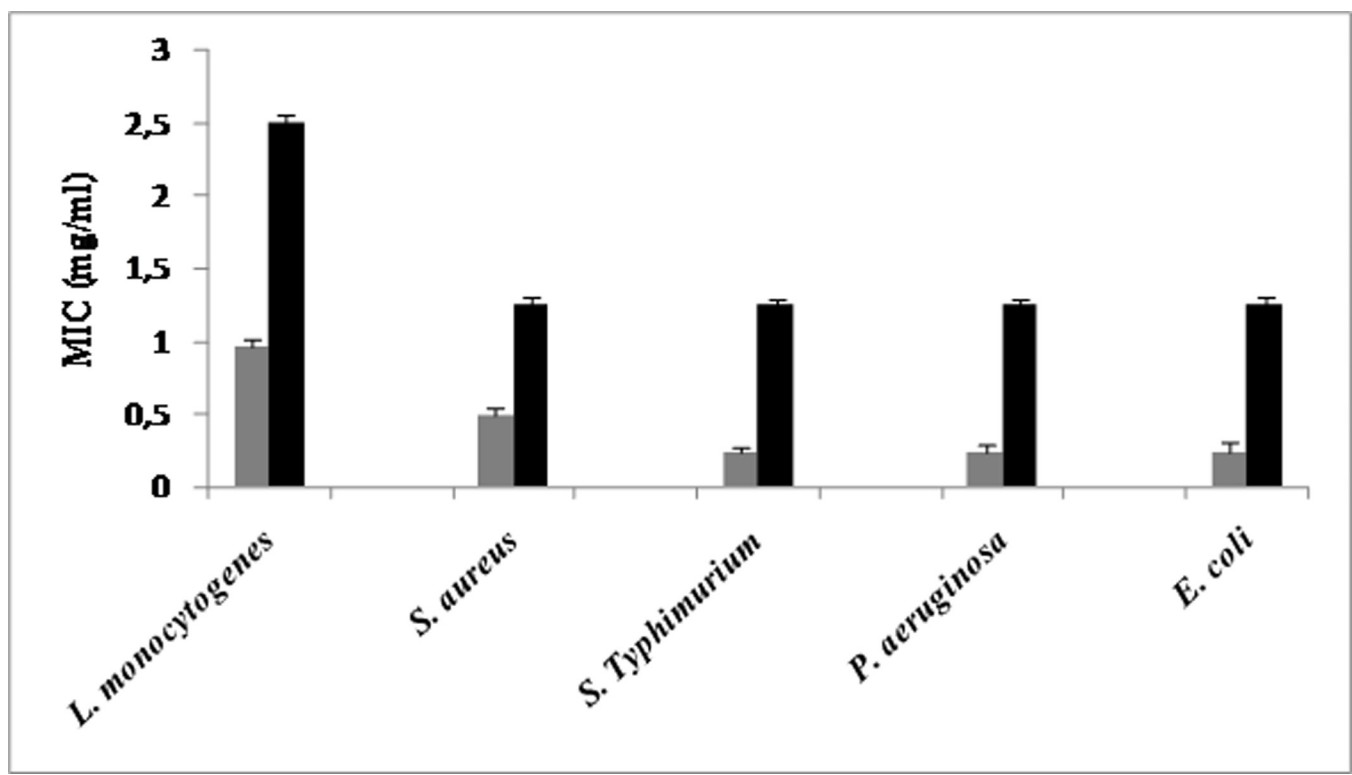

**Fig 1. Minimum inhibitory concentration of IP6 (□) and CA (□) against indicator strains (*L. monocytogenes* ATCC 19117, *S. aureus* ATCC 6538, *S.* Typhimurium ATCC 14028, *P. aeruginosa* ATCC 49189 and *E. coli* ATCC 8739.).** Values represent the means of triplicate experiments with comparable results.

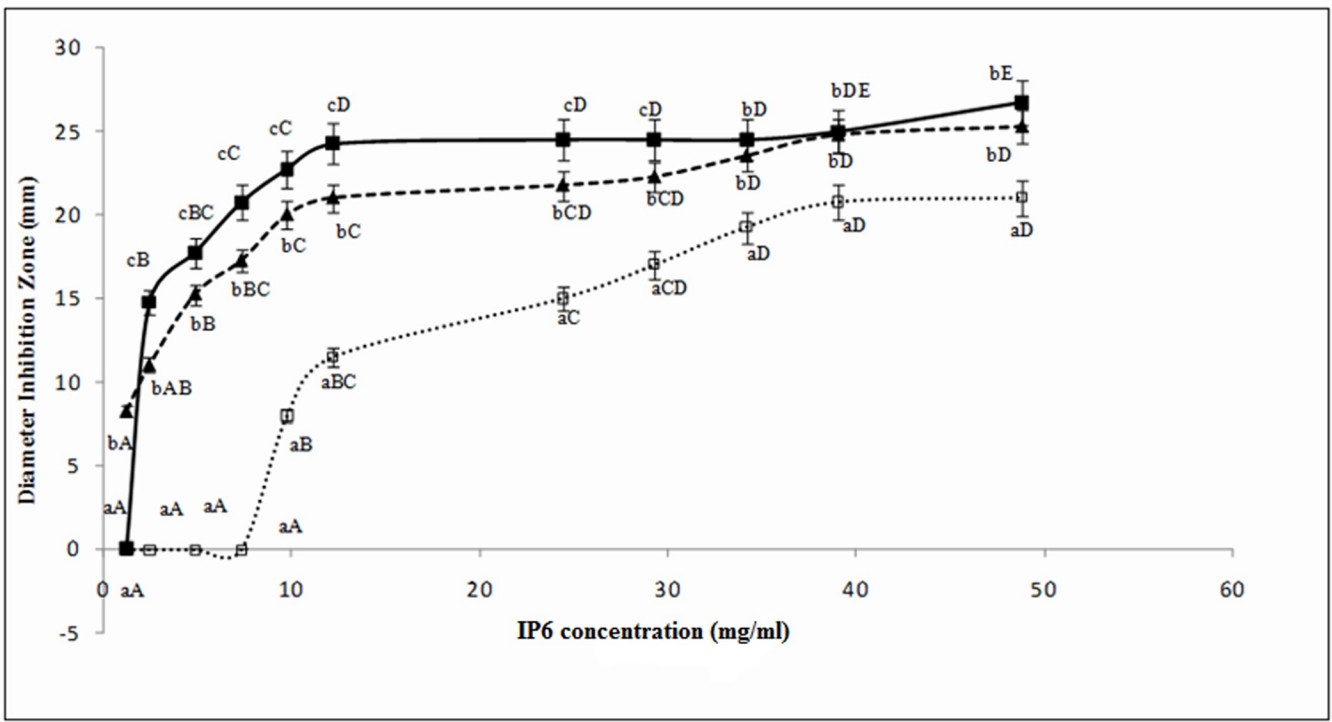

**Fig 2. Diameter inhibition zone of IP6 against _L. monocytogenes_ ATCC 19117(□), _S. aureus_ ATCC 6538 (□) and _S._ Typhimurium ATCC 14028(↘).** Values represent the means of triplicate experiments with comparable results.

**3.1.2. Diameter inhibition assay of phytic acid.** The results of the diameter inhibition assays that assessed the antibacterial effects of IP6 are shown in Fig 2. For all of the tested concentrations, this natural compound had an antagonistic effect on all indicator strains tested. The diameter of the inhibition zones generally increased with higher IP6 concentrations ($P$ <0.05). Furthermore, IP6 showed a strong inhibitory ($P<0.05$) effect and varying degrees of antibacterial activity against all of the strains tested, and the inhibition zones were in the range of 8.25–26.75 mm. At a concentration of 50 mg/ml, the largest diameters were recorded for _S. aureus_ and _S._ Typhimurium while _L. monocytogenes_ exhibited the smallest diameter. In general, the inhibition zones evolved exponentially ($P<0.05$) and, then, not significantly ($P>0.05$) between 34.16 and 50 mg/ml.

Discs containing 50 mg/ml of IP6 showed the largest mean inhibition zone diameter against _S. aureus_, while the discs containing 0.25 mg/ml showed the smallest diameter. The inhibition zones recorded for IP6 concentrations between 12.5 and 40 mg/ml were 24.5 mm in diameter. As the diameter remained the same ($P>0.05$) this clearly demonstrates that IP6 at 12.5 mg/ml brought about a significant ($P<0.05$) reduction in terms of the _S. aureus_ counts. We also found that the inhibitory action of IP6 against _S._ Typhimurium ATCC 14028 was comparable to that of _S. aureus_ ATCC 6538. Indeed, this inhibition started from a low concentration of IP6 (1.22 mg/ml) and stabilized ($P>0.05$) at 34.16 mg/ml. However, for _L. monocytogenes_ ATCC 19117, the antagonistic activity of IP6 was only observed from a concentration of 10 mg/ml.

Values with a different letter (a–c) of a same IP6 concentration are significantly different ($P< 0.05$).

Values with a different letter (A–E) of a same diameter inhibition zone are significantly different ($P<0.05$)

### 3.2. Mode of action of phytic acid

**3.2.1. Effect of the dose of phytic acid on the growth of *L. monocytogenes*, *S. aureus* and *S.* Typhimurium *in vitro* using a linear model (ANOVA).** To investigate the effects of varying IP6 dose on *L. monocytogenes*, *S. aureus* and *S.* Typhimurium, the bacterial growth was followed over 24 h and evaluated in comparison with the control culture (without addition of IP6), following the protocol developed by Jiang et al. [45]. IP6 was added to *L. monocytogenes* ATCC 19117, *S. aureus* ATCC 6538 and *S.* Typhimurium ATCC 14028 cells after 3 h of incubation, when growth reached the beginning of the exponential phase (cell density of about $10^6$ CFU/ml). **Fig 3A, 3B and 3C** shows that when 1×MIC, 2×MIC and 4×MIC of IP6 were added, a significant downward trend ($P<0.05$) in the viable count was observed and to some extent was dose-dependent. These results indicate that IP6 has a bactericidal activity against *L. monocytogenes* ATCC 19117, *S. aureus* ATCC 6538 and *S.* Typhimurium ATCC 14028. In fact, for *L. monocytogenes* ATCC 19117 and *S. aureus* ATCC 6538, a rapid killing action occurred one hour after the addition of 1×MIC, 2×MIC and 4×MIC of IP6 (incubation time of 4 h) with 1.2, 1.45, 4.29 and 4.57 $\log_{10}$ reductions in the density of the cells. The inhibition of growth even persisted 21 h after the addition of IP6. For the control culture, we noted an increase in the number of viable cells cultured in the absence of IP6 at 24 h of incubation ($10^{13}$ CFU/ml for *L. monocytogenes* ATCC 19117, $10^9$ CFU/ml for *S. aureus* ATCC 6538 and $10^7$ CFU/ml for *S.* Typhimurium ATCC 14028). Interestingly, the early logarithmic growth phase of *S.*

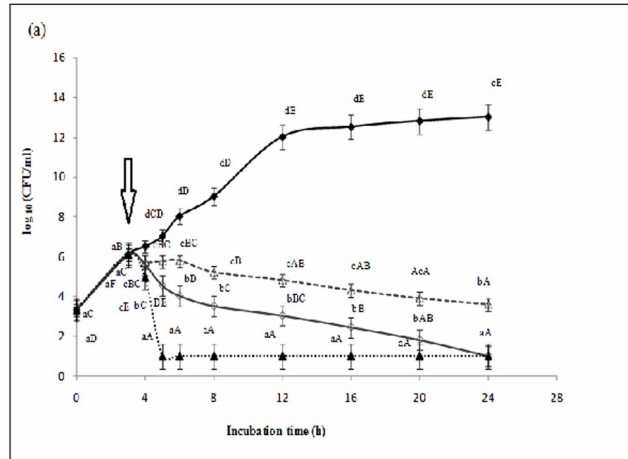
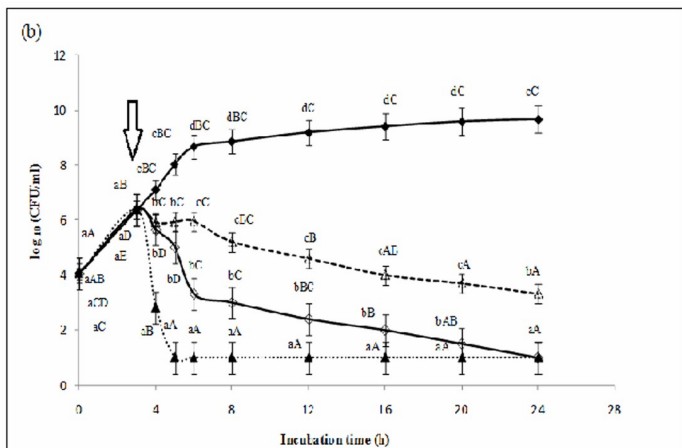
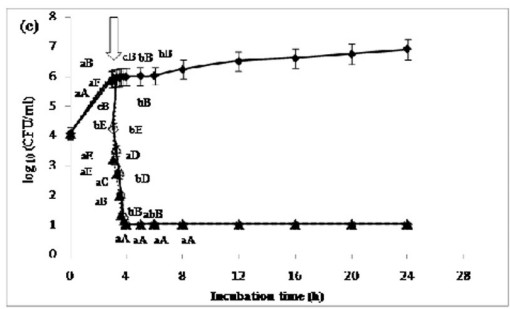

**Fig 3.** Influence of the dose of IP6 on the growth of *L. monocytogenes* ATCC 19117 (**a**), *S. aureus* ATCC 6538(**b**) and *S.* Typhimurium ATCC 14028 (**c**) *in vitro* using a linear model (ANOVA).

**Table 1. *L. monocytogenes* ATCC 19117, *S. aureus* ATCC 6538 and *S.* Typhimurium ATCC 14028 behavior estimates of treatment (Trial) fixed effects.**

| Parameter | Estimate | Std. Error | df | T | Sig. | LowerBound | LowerBound |
|---|---|---|---|---|---|---|---|
| | | | *Listeria monocytogenes* ATCC19117 | | | | |
| Intercept | 12.195532 | 1.836295 | 15.000 | 6.641 | 0.000 (***) | 8.281562 | 16.109502 |
| Hour 0 (time of IP6 addition) | -6.105627 | 2.596913 | 15.000 | -2.351 | 0.023 (*) | -11.640817 | -0.570438 |
| Hour 1 (post IP6 addition) | -5.254537 | 2.596913 | 15.000 | -2.023 | 0.029 (*) | -10.789727 | 0.280652 |
| Hour 2(post IP6 addition) | -2.815143 | 2.596913 | 15.000 | -1.084 | 0.044 (*) | -8.350332 | 2.720047 |
| Hour 3(post IP6 addition) | -2.103050 | 2.596913 | 15.000 | -0.810 | 0.131 (ns) | -7.638239 | 3.432139 |
| Hour 21(post IP6 addition) | 0[a] | 0 | . | . | . | . | . |
| Trial | -2.959106 | 0.553664 | 15.000 | -5.345 | 0.069 (ns) | -4.139213 | -1.779000 |
| Hour 0 × Trial | 2.959106 | 0.782999 | 15.000 | 3.779 | 0.002 (**) | 1.290184 | 4.628029 |
| Hour 1 × Trial | 2.443571 | 0.782999 | 15.000 | 3.121 | 0.017 (*) | 0.774649 | 4.112493 |
| Hour 2 × Trial | 0.984288 | 0.782999 | 15.000 | 1.257 | 0.027 (*) | -0.684635 | 2.653210 |
| Hour 3 × Trial | 0.781291 | 0.782999 | 15.000 | 0.998 | 0.134 (ns) | -0.887631 | 2.450214 |
| Hour 21 × Trial | 0[a] | 0 | . | . | . | . | . |
| | | | *Staphylococcus aureus* ATCC 6538 | | | | |
| Intercept | 9.402978 | 1.440860 | 15.000 | 6.526 | 0.000 (***) | 6.331858 | 12.474098 |
| Hour 0 | -3.028230 | 2.037683 | 15.000 | -1.486 | 0.031 (*) | -7.371449 | 1.314990 |
| Hour 1 | -0.928906 | 2.037683 | 15.000 | -0.456 | 0.049 (*) | -5.272125 | 3.414314 |
| Hour 2 | 0.995682 | 2.037683 | 15.000 | 0.489 | 0.249 (ns) | -3.347537 | 5.338902 |
| Hour 3 | 1.181910 | 2.037683 | 15.000 | 0.580 | 0.611(ns) | -3.161310 | 5.525129 |
| Hour 21 | 0[a] | 0 | . | . | . | . | . |
| Trial | -2.267845 | 1.058362 | 528.35 | -2.143 | 0.075 (ns) | -4.346960 | -0.188731 |
| Hour 0 × Trial | 2.267845 | 1.496750 | 528.35 | 1.515 | 0.013 (*) | -0.672467 | 5.208157 |
| Hour 1× Trial | 1.043681 | 1.496750 | 528.35 | 0.697 | 0.039 (*) | -1.896631 | 3.983994 |
| Hour 2 × Trial | 0.067139 | 1.496750 | 528.35 | 0.045 | 0.254 (ns) | -2.873173 | 3.007451 |
| Hour 3 × Trial | -0.063827 | 1.496750 | 528.35 | -0.043 | 0.566 (ns) | -3.004139 | 2.876485 |
| Hour 21 × Trial | 0[a] | 0 | . | . | . | . | . |
| | | | *Salmonella* Typhimurium ATCC 14028 | | | | |
| Intercept | 6.683287 | 2.171377 | 15.000 | 3.078 | 0.008 (**) | 2.055107 | 11.311466 |
| Hour 0 | -0.802473 | 3.070790 | 15.000 | -0.261 | 0.010 (*) | -7.347707 | 5.742761 |
| Hour 1 | -1.983238 | 3.070790 | 15.000 | -0.646 | 0.044 (*) | -8.528472 | 4.561996 |
| Hour 2 | -2.442463 | 3.070790 | 15.000 | -0.795 | 0.065 (ns) | -8.987697 | 4.102771 |
| Hour 3 | -2.425511 | 3.070790 | 15.000 | -0.790 | 0.249 (ns) | -8.970746 | 4.119723 |
| Hour 21 | 0[a] | 0 | . | . | . | . | . |
| Trial | -1.670822 | 8.797076 | 488978 | -0.190 | 0.104 (ns) | -18.912811 | 15.571168 |
| Hour 0 × Trial | 1.670822 | 12.44094 | 488978 | 0.134 | 0.016 (*) | -22.713034 | 26.054677 |
| Hour 1× Trial | 0.495809 | 12.44094 | 488978 | 0.040 | 0.033 (*) | -23.888046 | 24.879665 |
| Hour 2 × Trial | 0.610616 | 12.44094 | 488978 | 0.049 | 0.226 (ns) | -23.773240 | 24.994471 |
| Hour 3 × Trial | 0.606378 | 12.44094 | 488978 | 0.049 | 0.346 (ns) | -23.777478 | 24.990233 |
| Hour 21 × Trial | 0[a] | 0 | . | . | . | . | . |

[a]:This parameter is set to zero because it is redundant.

Std. Error: standard error; df: The degrees of freedom; t: the Student t-statistic; Sig.: the p-value (associated with the correlation).ns: $P > 0.05$;

*$P < 0.05$;

**$P < 0.01$;

***$P < 0.001$.

Typhimurium ATCC 14028 resulted in a rapid decrease in a period of less than one hour. It should be noted that this is the first time that it has been shown that IP6 alone can completely

prevent the growth of foodborne bacterial pathogens such as *L. monocytogenes*, *S. aureus* and *S.* Typhimurium. This stands in contrast to the study carried out by Bari et al. (2005), which showed that the combination of nisin (50 μg/ml)-IP6 (0.02%) and nisin-pediocin (100 AU/ml)-IP6 caused significant reductions in *L. monocytogenes* growth, but the total inactivation of the foodborne pathogen was not achieved by either the individual or the combined application of these antimicrobial agents. This could be due to the lower concentration of IP6 used in their study (0.02%) [46]. Taken together, our results indicate that IP6 exerts a dose-dependent bactericidal effect. According to Kim and Rhee [21], this effect might by due to the strong chelating capacity of IP6. Indeed, IP6 contains six reactive phosphate groups, which are responsible for its strong chelating capacity [21, 47].

Viable cell counts $\log_{10}$ (CFU/ml) in the absence (Control: ) and in the presence of 1×MIC (△), 2×MIC (◇) and 4×MIC (▲) of IP6. The time of addition of IP6 (incubation time of 3 h) is indicated by an arrow.

Values represent the means of triplicate experiments with comparable results.

Values with a different letter (a–d) of a same incubation time are significantly different (*P*< 0.05).

Values with a different letter (A–F) of a same IP6 dose are significantly different (*P*< 0.05)

**3.2.2 Effect of the dose of phytic acid on the growth of *L. monocytogenes*, *S. aureus* and *S.* Typhimurium *in vitro* using a general linear model (ANCOVA).** Analysis of covariance (ANCOVA) is a general linear model (GLM) that combines ANOVA with linear regression [48]. Descriptive statistics of the mixed model for the time-related survival of *L. monocytogenes*, *S. aureus* and *S.* Typhimurium(in PALCAM, Chapman and XLD broths, respectively) following treatment with various concentrations of IP6 are presented in **Tables 1** and **2**. The tests of fixed effects have an ANOVA-style test for each fixed effect in the model. This means that a single overall test can be used to assess the usefulness of a given explanatory variable, without focusing on individual levels. Explanatory variables that do not have a significant fixed effect can be removed and then the mixed effect analysis can be rerun using a simpler model with fewer explanatory variables (see **Tables 1** and **3**). ANCOVA was used to examine the differences in the means of the dependent variables. The independent variables were the six sampling times (0, 1, 2, 3 and 21 h post IP6 addition), and the four treatments (Trial 1: Control samples, Trial 2: 1×MIC, Trial 3: 2×MIC and Trial 4: 4×MIC). As shown in **Table 1,** for *L. monocytogenes*, a significant effect (*P*<0.05) was found at 0 (*P* = 0.023), 1 (*P* = 0.029) and 2 h (*P* = 0.044). In contrast, for *S. aureus*, and *S.* Typhimurium the most significant *P* values were observed at 0 h (*P* = 0.031 and P = 0.010, respectively) and 1 hour (*P* = 0.049 and P = 0.044, respectively). A significant interaction (*P*<0.05) between the treatments (all trials) and the bacterial growth time was

**Table 2. Estimates of covariance parameters in *L. monocytogenes* ATCC 19117, *S. aureus* ATCC 6538 and *S.* Typhimurium ATCC 14028 behavior estimates of treatment (Trial) fixed effects.**

| Parameter | Estimate | Std. Error | Wald Z | *P*. | LowerBound | LowerBound |
|---|---|---|---|---|---|---|
| *Listeria monocytogenes* ATCC19117 | | | | | | |
| Residual | 3.065435 | 1.119339 | 2.739 | 0.006 | 1.498571 | 6.270570 |
| TRIAL [subject = ID] Variance | 0.696690 | 23726566.4060 | 0.000 | 0.007 | 0.000000 | . |
| *Staphylococcus aureus* ATCC 6538 | | | | | | |
| Residual | 1.887343 | .689160 | 2.739 | 0.006 | .922648 | 3.860696 |
| TRIAL [subject = ID] Variance | 17.194269 | 7.59250110 | 0.000 | 0.019 | 0.000000 | . |
| *Salmonella* Typhimurium ATCC 14028 | | | | | | |
| Residual | 4.286251 | 1.565118 | 2.739 | 0.006 | 2.095380 | 8.767837 |
| TRIAL [subject = ID] Variance | 12.54789 | 257.15489 | 0.000 | 0.012 | 0.000000 | . |

**Table 3. *L. monocytogenes* ATCC 19117, *S. aureus* ATCC 6538 and *S.* Typhimurium ATCC 14028 behavior estimates of incubation time (h) fixed effects.**

| Parameter | Estimate | Std. Error | df | T | Sig. | LowerBound | LowerBound |
|---|---|---|---|---|---|---|---|
| | | | *Listeria monocytogenes* ATCC19117 | | | | |
| Intercept | 3,909020 | 0,957314 | 14,666 | 4,083 | 0.014 (*) | 1,864494 | 5,953546 |
| Trial 1 | 3,415564 | 1,218892 | 12,745 | 2,802 | 0.277 (ns) | 0,776946 | 6,054182 |
| Trial 2 | 3,015635 | 1,218892 | 12,745 | 2,474 | 0.122 (ns) | 0,377017 | 5,654253 |
| Trial 3 | 2,494664 | 1,218892 | 12,745 | 2,047 | 0.130 (ns) | -0,143954 | 5,133282 |
| Trial 4 | 0,172542 | 1,218892 | 12,745 | ,142 | 0.910 (ns) | -2,466076 | 2,811161 |
| Hour | -0,636441 | 0,342916 | 2,829 | -1,856 | 0.022 (*) | -1,766075 | 0,493193 |
| Trial 1 ×Hour | 0,485649 | 0,127775 | 12,745 | 3,801 | 0.021 (**) | 0,209046 | 0,762251 |
| Trial 2 ×Hour | 0,053229 | 0,127775 | 12,745 | ,417 | 0.008 (**) | -0,223374 | 0,329831 |
| Trial 3 ×Hour | -0,098872 | 0,127775 | 12,745 | -,774 | 0.006 (**) | -0,375474 | 0,177731 |
| Trial 4 ×Hour | -0,008869 | 0,127775 | 12,745 | -,069 | 0.000 (***) | -0,285471 | 0,267733 |
| | | | *Staphylococcus aureus* ATCC 6538 | | | | |
| Intercept | 3,330309 | 1,001639 | 14,862 | 3,325 | 0.004 (**) | 1,193637 | 5,466980 |
| Trial 1 | 4,751291 | 1,299773 | 12,677 | 3,655 | 0.071 (ns) | 1,936014 | 7,566569 |
| Trial 2 | 3,670235 | 1,299773 | 12,677 | 2,824 | 0.093 (ns) | ,854958 | 6,485513 |
| Trial 3 | 2,967218 | 1,299773 | 12,677 | 2,283 | 0.122 (ns) | ,151940 | 5,782495 |
| Trial 4 | 0,077913 | 1,299773 | 12,677 | 0,060 | 0.966 (ns) | -2,737364 | 2,893191 |
| Hour | -0,541623 | 0,307282 | 1,927 | -1,763 | 0.049 (*) | -1,913269 | ,830023 |
| Trial 1 ×Hour | 0,267524 | 0,136253 | 12,677 | 1,963 | 0.022 (*) | -,027597 | ,562645 |
| Trial 2 ×Hour | 0,007960 | 0,136253 | 12,677 | 0,058 | 0.009 (**) | -,287161 | ,303082 |
| Trial 3 ×Hour | -0,125313 | 0,136253 | 12,677 | -0,920 | 0.001 (**) | -,420434 | ,169808 |
| Trial 4 ×Hour | -0,004005 | 0,136253 | 12,677 | -0,029 | 0.000(***) | -,299126 | ,291116 |
| | | | *Salmonella* Typhimurium ATCC 14028 | | | | |
| Intercept | 4.968089 | 0.753077 | 12.138 | 6.597 | 0,000 (***) | 3.329336 | 6.606842 |
| Trial 1 | 3.683758 | 0.883156 | 11.714 | 4.171 | 0,001 (**) | 1.754296 | 5.613219 |
| Trial 2 | $3.90459 \times 10^{-15}$ | 0.883156 | 11.714 | 0.000 | 0.214 (ns) | -1.929461 | 1.929461 |
| Trial 3 | $3.94568 \times 10^{-15}$ | 0.883156 | 11.714 | 0.000 | 0.354 (ns) | -1.929461 | 1.929461 |
| Trial 4 | $3.94568 \times 10^{-15}$ | 0.883156 | 11.714 | 0.000 | 0.555 (ns) | -1.929461 | 1.929461 |
| Hour | -2.148155 | 0.927415 | 3.026 | -2.316 | 0,031 (*) | -5.085513 | .789204 |
| Trial 1 ×Hour | 0.238283 | 0.092580 | 11.714 | 2.574 | 0,0011(*) | .036020 | .440545 |
| Trial 2 ×Hour | $-2.7329 \times 10^{-16}$ | 0.092580 | 11.714 | 0.000 | 0.000 (***) | -.202263 | .202263 |
| Trial 3 ×Hour | $-2.8090 \times 10^{-15}$ | 0.092580 | 11.714 | 0.000 | 0.000 (***) | -.202263 | .202263 |
| Trial 4 ×Hour | $-2.8090 \times 10^{-16}$ | 0.092580 | 11.714 | 0.000 | 0.000 (***) | -.202263 | .202263 |

Std. Error: standard error; df: The degrees of freedom; t: the Student t-statistic; Sig.: the p-value (associated with the correlation).ns: $P>0.05$;

*$P<0.05$;

**$P<0.01$;

***$P<0.001$.

Trial 1: Control (No IP6 added); Trial 2: 1×MIC. Trial 3: 2×MIC. Trial 4: 4×MIC.

shown for 0, 1 and 2 h for *L. monocytogenes* and for 0 and 1 hour for *S. aureus* and *S.* Typhimurium (**Table 1**). The results shown in **Tables 1** and **2** indicate that the first h (0, 1 and 2) are especially important for bacterial growth inhibition. The first 2 h after adding IP6 would appear to be critical for the inhibition of *L. monocytogenes*, as after this period, no significant inhibition was found. For *S. aureus* and *S.* Typhimurium, growth was inhibited one hour after adding IP6.

The covariance parameters are presented in **Table 2**. The intercept variances were estimated as 2.554530, 1.572786 and 4.286251 for *L. monocytogenes*, *S. aureus* and *S.*

**Table 4. Estimates of covariance parameters in *L. monocytogenes* ATCC 19117. *S. aureus* ATCC 6538 and *S.* Typhimurium ATCC 14028 behavior estimates of incubation time (h) fixed effects.**

| Parameter | Estimate | Std. Error | Wald Z | P | LowerBound | LowerBound |
|---|---|---|---|---|---|---|
| *Listeria monocytogenes* ATCC19117 | | | | | | |
| Residual | 2.450288 | 0.935779 | 2.618 | 0.009 | 1.159147 | 5.179597 |
| HOUR [subject = ID] Variance | 0.085107 | 0.105096 | 0.810 | 0.041 | 0.007566 | 0.957389 |
| *Staphylococcus aureus* ATCC 6538 | | | | | | |
| Residual | 3.501211 | 1.107180 | 3.162 | 0.002 | 1.883843 | 6.507165 |
| HOUR [subject = ID] Variance | 32.26787 | 11.51787 | 0.000 | 0.031 | 0.000000 | . |
| *Salmonella* TyphimuriumATCC 14028 | | | | | | |
| Residual | 1,325082 | 0,547535 | 2,420 | 0,016 | 0,589548 | 2,978285 |
| TRIAL [subject = ID] Variance | 3,123839 | 2,784617 | 1,122 | 0,021 | 0,544403 | 17,924910 |

Typhimurium, respectively (**Table 2**). The null hypothesis for this parameter is a variance of zero, which would indicate that a random effect is not present. This can be assessed using a statistical test called a Wald Z statistic [49–51]. For *L. monocytogenes*, *S. aureus* and *S.* Typhimurium, this test was run and the null hypothesis (Wald Z = 0.000, $P$ = 0.019), (Wald Z = 0.000, $P$ = 0.007) and (Wald Z = 0.000, $P$ = 0.012) was rejected. This suggests that there are important unmeasured explanatory variables that affect the results in a way that appears random because we do not know the values of the missing explanatory variables (**Table 2**).

The interaction estimates for the differences in slope between trial 4 (4×MIC) and the other trials are shown in Table 2A for the three strains. It is very important to note that the parameter estimates given in the fixed effects are estimates of mean parameters. The effects of the treatments, the bacterial growth time and their interaction on the inhibition of *L. monocytogenes*, *S. aureus* and *S.* Typhimurium are shown in **Table 3**. A significant interaction ($P<0.05$) was found between all of the treatments and the bacterial growth time. This was highly significant ($P<0.001$) for the interaction between trial 4 (4×MIC) and time for *L. monocytogenes*, *S. aureus* and *S.* Typhimurium(in PALCAM, Chapman and XLD media, respectively) (**Table 3**). These results show that high concentrations of IP6 coupled with incubation time actively inhibit the studied bacteria. Interestingly, for *S.* Typhimurium, 1×MIC of IP6 is sufficient for bacterial inhibition.

As presented in **Table 4**, for *L. monocytogenes*, *S. aureus* and *S.* Typhimurium, the intercept variances ((Wald Z = 0.810, $P$ = 0.041), (Wald Z = 0.000, $P$ = 0.031) and (Wald Z = 0.000, $P$ = 0.021)) were found to be greater than zero (**Table 4**). The results shown in **Tables 3 and 4** show that there are interactions between the trial and incubation time for *L. monocytogenes*, *S. aureus* and *S.* Typhimurium.

## 4. Conclusion

In this study, we demonstrated for the first time the strong inhibitory effect of phytic acid (IP6) against the proliferation of both Gram-positive and Gram-negative foodborne pathogenic bacteria. This was achieved by determining the MIC values and diameter inhibition assays. Of note, IP6 was found to be especially effective against Gram-negative bacteria and more efficient than citric acid, which is widely used as a natural sanitizer in the food industry to reduce the growth rate of foodborne pathogenic bacteria. The analysis study assessing the mode of action confirms these results and indicates that IP6 exerts a dose-dependent bactericidal effect against *L. monocytogenes*, *S. aureus* and *S.* Typhimurium.

Taken together, our results reinforce suggestions that IP6, a natural and biodegradable GRAS compound, could be used for the development of future environmentally friendly applications in the food industry.

## Supporting information

**S1 Data.**
(PDF)

## Acknowledgments

This work was supported by the Tunisian Government (Contract Program LMB-CBS, 2015–2018) and by the University of Jeddah (Saudi Arabia) through the International Cooperation Grant Research Project N° UJ-02-009-ICGR.

The authors acknowledge Prof. Pierfrancesco Cerruti (Institute for Polymers, Composites and Biomaterials, Pozzuoli–Italy) for critical reading of the manuscript and collaboration.

## Author Contributions

**Conceptualization:** Hichem Chouayekh.

**Data curation:** Ines Boukhris.

**Formal analysis:** Ines Boukhris, Slim Smaoui, Karim Ennouri, Hichem Chouayekh.

**Funding acquisition:** Othman A. Alghamdi.

**Investigation:** Ines Boukhris.

**Methodology:** Ines Boukhris, Slim Smaoui, Nawres Morjene.

**Project administration:** Hichem Chouayekh.

**Resources:** Hichem Chouayekh.

**Software:** Slim Smaoui, Karim Ennouri.

**Supervision:** Slim Smaoui.

**Validation:** Ines Boukhris, Slim Smaoui.

**Visualization:** Ines Boukhris, Slim Smaoui.

**Writing – original draft:** Ines Boukhris, Slim Smaoui, Karim Ennouri, Ameny Farhat-Khemakhem, Monia Blibech, Othman A. Alghamdi.

**Writing – review & editing:** Ines Boukhris, Slim Smaoui, Ameny Farhat-Khemakhem, Monia Blibech, Othman A. Alghamdi, Hichem Chouayekh.

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
