## [Decision Letter · Decision Letter 0]

26 Feb 2020

PONE-D-19-34291

Towards understanding of the antagonistic activity of phytic acid against common foodborne bacterial pathogens using a general linear model

PLOS ONE

Dear Hichem Chouayekh,

Thank you for submitting your manuscript to PLOS ONE. After careful consideration, we feel that it has merit but does not fully meet PLOS ONE’s publication criteria as it currently stands. Therefore, we invite you to submit a revised version of the manuscript that addresses the points raised during the review process.

ACADEMIC EDITOR: 

The materials and methods must be carefully checked and improved as reported by both the reviewers. The affermations/sentence reported in the introduction must be supported by references. 

We would appreciate receiving your revised manuscript by 26 March 2020. To enhance the reproducibility of your results, we recommend that if applicable you deposit your laboratory protocols in protocols.io, where a protocol can be assigned its own identifier (DOI) such that it can be cited independently in the future. For instructions see: http://journals.plos.org/plosone/s/submission-guidelines#loc-laboratory-protocols

We look forward to receiving your revised manuscript.

Kind regards,

Filippo Giarratana

Academic Editor

PLOS ONE

Journal Requirements:

"This work was supported by the Tunisian Government (Contract Program LMB-CBS, 2015- 2018) and the University of Jeddah (Saudi Arabia) through the International Cooperation Grant Research Project N° UJ-02-009-ICGR. ..."

"The authors received no specific funding for this work"

Reviewers' comments:

Reviewer's Responses to Questions

**Comments to the Author**

1. Is the manuscript technically sound, and do the data support the conclusions?

Reviewer #1: Partly

Reviewer #2: Yes

2. Has the statistical analysis been performed appropriately and rigorously? 

Reviewer #1: I Don't Know

Reviewer #2: N/A

3. Have the authors made all data underlying the findings in their manuscript fully available?

Reviewer #1: Yes

Reviewer #2: Yes

4. Is the manuscript presented in an intelligible fashion and written in standard English?

Reviewer #1: Yes

Reviewer #2: Yes

5. Review Comments to the Author

Reviewer #1: In this study, the authors describe the antagonistic activity of phytic acid against common foodborne bacterial pathogens using a general linear model. The results obtained in this study showed an interesting effect of IP6 against the tested bacteria. Unfortunately, the methods used to perform this work are primitive. However, I recommend adding the following tests in order to improve the quality of this work:

- Studying the kinetic parameters of microbial growth in the presence of IP6 (PLoS ONE 10(2): e0114026. doi: 10.1371/journal. pone.0114026).

- Studying the antimicrobial activity of IP6 in a food model.

- The determination of the mode of action of IP6 on the tested bacteria should be reinforced by others test, like Proton Motive Force (PMF) (to check the effect of IP6 on membrane integrity), Scanning Electron Microscopy (SEM)….( BioMed Research Internationa, Doi: 10.1155/2017/7657190).

- After adding these tests, the title can be changed to: “in vitro and in situ effects of phytic acid on the growth of common foodborne bacterial pathogens”.

In addition, the manuscript writing should be improved:

- Salmonella Typhimurium must be written as Salmonella in the italic form with a capital letter and Typhimurium with the non-italic form with a capital letter.

- Lines no. 75-78: “For this reason, in November 2017, the World…… of antibiotics for human medicine”; the authors talk about the recommendation of WHO published in 2017, but the reference at the end of this sentence was published in 2016 and does not presented these recommendations. So, can the authors check again this reference?

- In materiel and method (2.3. Determination of minimum inhibitory concentrations): Can the authors specify the culture media (name and volume) used for the growth of bacteria in microplates.

- In materiel and method (2.4.Agar diffusion method): the reference Smaoui et al. (2010) (43) does not describe the protocol of the agar diffusion method but they referred to another reference. So, can the authors describe the protocol in detail or put another reference that describes it.

- In materiel and method (2.5. Mode of action of phytic acid): To study the mode of action, authors should not use selective media (PALCAM, CHAMPAN and XLD), because these media contain selective substances that may act in synergic/inhibitory/ antagonistic with the tested compound. So, I recommend using TSB or TSA, TSYEB or TSYEA, Mueller Hinton agar or Mueller Hinton broth…to study the mode of action of IP6.

- The resolution of Figures should be improved.

Reviewer #2: The study presented by Boukhris et al. investigates the antibacterial efficiency of phytic acid against common food associated pathogens.

Since alternative methods to control the all over food quality and consumer safety become more important against the background of antimicrobial resistance and the demand of consumers for healthy, natural foods the subject is of interest to a wide number of readers.

The work was generally well prepared and the manuscript is generally well written and clear to read. Nevertheless, from my point of view the manuscript needs some basical revisions prior to publication.

General comments:

The authors should check the whole document whether all the statements made are sufficiently and accurately referenced and if they are formatted according to the requirements of the journal. The whole documents should be double-spaced (e.g. L221-228). Please uniformize the units (e.g. hours or h) throughout the whole document. Check the text for correct formatting and spaces (e.g. P and P and spaces before and after =)

Please uniformize the row heights and text alignment in all tables.

LL 6-7: Please state which authors contributed equally to this study

L10+12 Tunisie/Tunisia please uniformize

L7: Is the superscript „3“ in formatted in bold?

L 38: Please give the full name of the PALCAM agar as already done for XLD broth

LL 50-51: Please add a reference.

LL60-61: Please add a reference and also check with the sentence in LL 50-51.

L 60: „produces“ as an alternative: leads to / causes.

L 64-65: Please add a reference.

L 73: Could you please check whether this the right reference, since you are referring to a single study and not to review?

LL 73-75: Please add a reference

L 78: Reference 22: The reference doesn’t seem to match. Please cite the original recommendation of the WHO.

LL 83-85: Please add a reference.

LL: 91-93: Please add a reference

LL 98-99: Please add a reference

LL 102-103: Please adapt the output style to the requirements of tthe journal.

LL 110: inhibition diameters: in which test?

LL 113-117: I understand why you are pointing at this, but especially the last sentence confuses me. Wouldn’t it be easier to skip the whole paragraph and just mention IP6 an citric acid in the text/description of the respective method?

Material and Methods:

L120: remove the comma after ATCC 19117

L122: remove the comma after ATCC 49189

L 125: Chapman medium (Oxoid… please complete if mentioned for the first time

L 129 (CFU)/ml

LL 129-130: Please adapt the outputstyle to the requirements of the journal and remove the point after (2014)

LL 135: Please adapt the outputstyle to the requirements of the journal

LL 145-149: It doesn’t come to me easily how exactly the AWDA was performed. Please provide further details

L 152: Please adapt the outputstyle to the requirements of the journal

L 159: CFU per …?

L 182: measuring the inhibition zones in AWDA?

L 203: indicative strains- indicator strains?

LL 259-262: To my opinion, the data do not prove chelat forming. Perhaps you could rebuilt the sentence as: According to Kimm and Rhee this effect might by due to the strong chelating …. ?

LL 274 The ANCOVA and later on WALD Z statistic was not mentioned in the material and method section.

L285-286: I don’t think, that you have to explain the meaning and interpretation of P.

LL 313-315 Well, yes - but this sentence is confusing.

6. PLOS authors have the option to publish the peer review history of their article (what does this mean?). If published, this will include your full peer review and any attached files.

Reviewer #1: Yes: Abdelaziz ED-DRA

Reviewer #2: No

---

## [Author Response · Author response to Decision Letter 0]

18 Mar 2020

Response to Referees: Manuscript ID: PONE-D-19-34291

Title:

“Towards understanding the antagonistic activity of phytic acid against common foodborne bacterial pathogens using a general linear model”

By

Boukhris et al.,

Journal: PLOS ONE

Point-by-point answer to Reviewers comments

First of all, we would like to thank Reviewers for their interesting and constructive comments.

Required modifications are now addressed in all sections of the revised manuscript and are illustrated in red colour. 

Answers to the Reviewer 1 Comments

• In this study, the authors describe the antagonistic activity of phytic acid against common foodborne bacterial pathogens using a general linear model. The results obtained in this study showed an interesting effect of IP6 against the tested bacteria. 

Dear Reviewer, thank you very much for your kind judgment of our manuscript. We are grateful for the time and energy you expended to improve our work. In the following sections, you will find our responses to each of your points and suggestions. 

• Unfortunately, the methods used to perform this work are primitive. However, I recommend adding the following tests in order to improve the quality of this work:

- Studying the kinetic parameters of microbial growth in the presence of IP6 (PLoS ONE 10(2): e0114026. doi: 10.1371/journal. pone.0114026).

- Studying the antimicrobial activity of IP6 in a food model.

- The determination of the mode of action of IP6 on the tested bacteria should be reinforced by others test, like Proton Motive Force (PMF) (to check the effect of IP6 on membrane integrity), Scanning Electron Microscopy (SEM)….( BioMed Research Internationa, Doi: 10.1155/2017/7657190).

- After adding these tests, the title can be changed to: “in vitro and in situ effects of phytic acid on the growth of common foodborne bacterial pathogens”.

Dear Reviewer, thank you very much for these pertinent suggestions. In fact, our study focused on the understanding of the mechanism of natural antimicrobial action of phytic acid (IP6), by the development of predictive mathematical models. To elucidate a better understanding on the phytic acid as an antibacterial agent, linear (ANOVA) and general (ANCOVA) models were mathematically used to model the growth rate of Listeria monocytogenes ATCC 19117, Staphylococcus aureus ATCC 6538 and Salmonella Typhimurium ATCC 14028. Furthermore, the combination of ANOVA analysis and ANCOVA approach was used for linking all data. It should be noted that predictive inactivation models have been developed in liquid laboratory media that can mimic the microbial environment.

In this regard, to the best of our knowledge, no data are available on simultaneously study of inactivation of three food borne pathogens (L. monocytogenes, S. aureus and S. Typhimurium) by predictive mathematical models.

- On the other hand, studying the antimicrobial activity of IP6 in a food model is currently in progress. Indeed, we recently investigated the effects of different concentrations of IP6 on shelf-life, microbiological, physicochemical and sensory qualities of stored raw beef and chicken meat products. The analysis of these results is ongoing and will be the subject of another publication. 

- Equally, for a conclusive confirmation of the mode of action of IP6 on the all tested bacteria, an experimental validation by Scanning Electron Microscopy (SEM) is currently in progress due to the unavailability of functional scanning electron microscope in region last year.

• In addition, the manuscript writing should be improved:

- Salmonella Typhimurium must be written as Salmonella in the italic form with a capital letter and Typhimurium with the non-italic form with a capital letter.

Dear Reviewer, according to your remarks; corrections are made in the text. 

Please see the revised version

- Lines no. 75-78: “For this reason, in November 2017, the World…… of antibiotics for human medicine”; the authors talk about the recommendation of WHO published in 2017, but the reference at the end of this sentence was published in 2016 and does not presented these recommendations. So, can the authors check again this reference?

Dear Reviewer, you are quite right. According to your remark changes are made as follows:

• For this reason, the World Health Organization (WHO) recommended that breeders and the food industry stop misusing antibiotics, such as for promoting animal growth and preventing disease in healthy animals, in order to preserve the effectiveness of antibiotics for human medicine [22].

The reference 

22. Holmes AH, Moore LS, Sundsfjord A, Steinbakk M, Regmi S, Karkey A, et al. Understanding the mechanisms and drivers of antimicrobial resistance. Lancet. 2016;387: 176-187. Was change by: 

22. World Health Organization (WHO). Strategic and Technical Advisory Group on Antimicrobial Resistance. Report of the first meeting Geneva, Sept 19–20, 2013; http://www.who.int/drugresistance/stag/amr_stag_meeting report 0913.pdf (accessed July 14, 2015).

Please see the revised version

- In materiel and method (2.3. Determination of minimum inhibitory concentrations): Can the authors specify the culture media (name and volume) used for the growth of bacteria in microplates.

Dear Reviewer, thank you for this interesting observation, as mentioned above, the correction was made and the following paragraphs were added to the section 2.3. Determination of minimum inhibitory concentrations in Materials and Methods part: 

The test was performed in sterile 96-well microplates with a final volume of 100 μl per well. A commercial stock solution of IP6 (50% (w/w) in water) and a stock solution of CA at 50% (w/w) in water were used. Then, the corresponding concentrations of IP6 and CA were transferred to each successive well in order to obtain a two-fold serial dilution of the original sample. In fact, each sample was dissolved to a final concentration of 0.078, 0.156, 0.312, 0.625, 1.25, 2.5, 5, 10 and 20 mg/mL and then filtered through 0.22 μm pore-size black polycarbonate filters (Millipore). To each test well 10 μl of cell suspension were added to final inoculum concentration of 106 CFU/ml of bacterium. Positive growth control well consisted of Listeria monocytogenes ATCC 19117, Staphylococcus aureus ATCC 6538 and Salmonella Typhimurium ATCC 14028 respectively growth in PALCAM, Chapman and XLD. Plates were then covered with the sterile plate covers and incubated at 37 °C for 24 h. As an indicator of microorganism growth, 25 μl of thiazolyl blue tetrazolium bromide (MTT) indicator solution (0.5 mg/ml) dissolved in sterile water was added to the wells and incubated at 37 °C for 30 min. The colourless tetrazolium salt acts as an electron acceptor and was reduced to a red-coloured formazan product by biologically active organisms. Where microbial growth was inhibited, the solution in the well remained clear after incubation with MTT. The determination of MIC values was done in triplicate.

Please see the revised version

- In materiel and method (2.4.Agar diffusion method): the reference Smaoui et al. (2010) (43) does not describe the protocol of the agar diffusion method but they referred to another reference. So, can the authors describe the protocol in detail or put another reference that describes it.

Dear Reviewer, thank you for this pertinent remark, 

• This reference: [43]. Smaoui S, Elleuch L, Bejar W, Karray-Rebai I, Ayadi I, Jaouadi B et al. Inhibition of fungi and gram-negative bacteria by bacteriocin BacTN635 produced by Lactobacillus plantarum sp. TN635. App Biochem Biotechnol. 2010; 162: 1132-1146 was changed by : 

• 44. Valgas C, Souza SMD, Smânia EF, Smânia Jr A. Screening methods to determine antibacterial activity of natural products. Braz. J. Microbiol. 2007; 38: 369-380.

Please see the revised version

- In materiel and method (2.5. Mode of action of phytic acid): To study the mode of action, authors should not use selective media (PALCAM, CHAMPAN and XLD), because these media contain selective substances that may act in synergic/inhibitory/ antagonistic with the tested compound. So, I recommend using TSB or TSA, TSYEB or TSYEA, Mueller Hinton agar or Mueller Hinton broth…to study the mode of action of IP6.

Thank you for this interesting remark. In fact, we have chosen to use only the corresponding selective media (PALCAM, Chapman and XLD) for cultivation of Listeria monocytogenes ATCC 19117, Staphylococcus aureus ATCC 6538 and Salmonella Typhimurium ATCC 14028 to track the exact growth of each bacterium. Equally, we avoided the TSB since it is a Non selective medium. 

- The resolution of Figures should be improved.

The Reviewer has perfectly reason for this comment. This comment has been addressed in the revised manuscript.

Please see the revised version

Answers to the Reviewer 2 Comments

The study presented by Boukhris et al. investigates the antibacterial efficiency of phytic acid against common food associated pathogens.

Since alternative methods to control the all over food quality and consumer safety become more important against the background of antimicrobial resistance and the demand of consumers for healthy, natural foods the subject is of interest to a wide number of readers.

The work was generally well prepared and the manuscript is generally well written and clear to read. Nevertheless, from my point of view the manuscript needs some basical revisions prior to publication.

 Dear Reviewer, thank you very much for your kind judgment about our paper. We are grateful for the time and energy you expended on our work. In the following sections, you will find our responses to each of your points and suggestions. 

General comments:

• The authors should check the whole document whether all the statements made are sufficiently and accurately referenced and if they are formatted according to the requirements of the journal. The whole documents should be double-spaced (e.g. L221-228). Please uniformize the units (e.g. hours or h) throughout the whole document. Check the text for correct formatting and spaces (e.g. P and P and spaces before and after =)

• Please uniformize the row heights and text alignment in all tables.

• LL 6-7: Please state which authors contributed equally to this study

• L7: Is the superscript „3“ in formatted in bold?

• L10+12 Tunisie/Tunisia please uniformize

Dear Reviewer, thank you for these interesting remarks. As suggested, corrections have been addressed in the revised manuscript. There is no equal contribution for the authors to this study.

Please see the revised version

• L 38: Please give the full name of the PALCAM agar as already done for XLD broth

Dear reviewer, as recommended, the full name of the PALCAM agar has been introduced 

Please see the revised version

• LL 50-51: Please add a reference.

Dear Reviewer, as recommended, two references were introduced in L50-51

1. Havelaar AH, Kirk MD, Torgerson PR, Gibb HJ, Hald T, Lake RJ, et al. World Health Organization global estimates and regional comparisons of the burden of foodborne disease in 2010. PLoS Med. 2015; 12.

2. Bhaskar SV. Foodborne diseases—disease burden. In Food Safety in the 21st Century. Academic Press; 2017. pp. 1-10.

Please see the revised version

• LL60-61: Please add a reference and also check with the sentence in LL 50-51.

Dear Reviewer, as recommended, the following reference was introduced in L60-61

 10. Morach M, Käppeli N, Hochreutener M, Johler S, Julmi J, Stephan R, et al. Microarray based genetic profiling of Staphylococcus aureus isolated from abattoir byproducts of pork origin. PLoS One. 2019;14(9).

Please see the revised version

• L 60: „produces“ as an alternative: leads to / causes.

Dear Reviewer, as suggested, correction was made. 

Please see the revised version

• L 64-65: Please add a reference.

Dear Reviewer, as recommended, the following reference was introduced in L64-65

 17. Nilsson OR, Kari L, Steele-Mortimer O. Foodborne infection of mice with Salmonella Typhimurium. PLoS One. 2019;14(8).

Please see the revised version

• L 73: Could you please check whether this the right reference, since you are referring to a single study and not to review?

Dear Reviewer, thank you for this pertinent remark, 

• This reference: 21. Kim NH, Rhee MS. Phytic acid and sodium chloride show marked synergistic bactericidal effects against nonadapted and acid-adapted Escherichia coli O157: H7 strains, Appl Environ Microbiol. 2016;82:1040-1049, was changed by : 

• 21. Parish ME, Beuchat LR, Suslow TV, Harris LJ, Garrett EH, Farber JN, Busta FF. Methods to reduce/eliminate pathogens from fresh and fresh‐cut produce. Compr. Rev. Food Sci. Food Saf. 2003;2: 161-173.

Please see the revised version

• LL 73-75: Please add a reference

Dear Reviewer, as recommended, the following reference was introduced in L64-65

22. World Health Organization (WHO). Strategic and Technical Advisory Group on Antimicrobial Resistance. Report of the first meeting Geneva, Sept 19–20, 2013; http://www.who.int/drugresistance/stag/amr_stag_meeting report 0913.pdf (accessed July 14, 2015).

Please see the revised version

• L 78: Reference 22: The reference doesn’t seem to match. Please cite the original recommendation of the WHO.

Dear Reviewer, as recommended, the following reference was introduced in L64-65

22. World Health Organization (WHO). Strategic and Technical Advisory Group on Antimicrobial Resistance. Report of the first meeting Geneva, Sept 19–20, 2013; http://www.who.int/drugresistance/stag/amr_stag_meeting report 0913.pdf (accessed July 14, 2015).

Please see the revised version

• LL 83-85: Please add a reference.

Dear Reviewer, as recommended, the following reference was introduced in L64-65

 27. Lin CM, Sheu SR, Hsu SC, Tsai YH. Determination of bactericidal efficacy of essential oil extracted from orange peel on the food contact surfaces. Food Control. 2010; 21(12):1710-1715.

Please see the revised version

• LL: 91-93: Please add a reference

Dear Reviewer, as recommended, the following reference was introduced in L64-65

 29. Schlemmer U, Frolich W, Prieto RM, Grases F. Phytate in foods and significance for humans: food sources, intake, processing, bioavailability, protective role and analysis. Mol Nutr Food Res. 2009;53:S330-S375.

Please see the revised version

• LL 98-99: Please add a reference

Dear Reviewer, as recommended, the following reference was introduced in L64-65

 31. Bhowmik A, Ojha D, Goswami D, Das R, Chandra NS, Chatterjee TK, et al. Inositol hexa phosphoric acid (phytic acid), a nutraceuticals, attenuates iron-induced oxidative stress and alleviates liver injury in iron overloaded mice. 2017. Biomed Phramacother. 2017;87: 443-450.

Please see the revised version

• LL 102-103: Please adapt the output style to the requirements of the journal.

Dear Reviewer, thank you for this remark. As suggested, correction was made. 

Please see the revised version

• LL 110: inhibition diameters: in which test?

Dear Reviewer, thank you for your pertinent remark. The sentence ‘This is achieved by: (i) determining the minimum inhibitory concentration (MIC) as compared to citric acid (CA); (ii) measuring the inhibition diameters, and (iii) illustrating the mode of action of IP6 for inhibiting pathogen growth., was replaced by:

This is achieved by: (i) determining the minimum inhibitory concentration (MIC) as compared to citric acid (CA); (ii) measuring the inhibition diameters of each indicator bacteria growth inhibition or reduction, and (iii) illustrating the mode of action of IP6 for inhibiting pathogen growth.

Please see the revised version

• LL 113-117: I understand why you are pointing at this, but especially the last sentence confuses me. Wouldn’t it be easier to skip the whole paragraph and just mention IP6 and citric acid in the text/description of the respective method?

Dear Reviewer, you are quite right, the correction was made as proposed, and the sentence’ All of the other chemicals used in this study were commercially available in analytical grade’ was deleted 

Material and Methods:

• L120: remove the comma after ATCC 19117

• L122: remove the comma after ATCC 49189

Dear Reviewer, the correction was made as proposed. 

Please see the revised version.

• L 125: Chapman medium (Oxoid… please complete if mentioned for the first time

Dear Reviewer, the correction was made as proposed: …Chapman medium (Oxoid, Basingstoke, Hampshire, UK)… 

Please see the revised version.

• L 129 (CFU)/ml

Dear Reviewer, the correction was made as proposed: ….106 colony-forming units of bacteria per milliliter (CFU/ml)…

Please see the revised version.

• LL 129-130: Please adapt the output style to the requirements of the journal and remove the point after (2014)

Dear Reviewer, the correction was made as proposed. 

Please see the revised version.

• LL 135: Please adapt the output style to the requirements of the journal

Dear Reviewer, the correction was made as proposed. 

Please see the revised version.

• LL 145-149: It doesn’t come to me easily how exactly the AWDA was performed. Please provide further details

Dear Reviewer, thank you for this pertinent comment. The following paragraph was added to the ‘2.4. Agar diffusion method’ section in the Materials and Methods part of the revised Manuscript. According to reviewer Comment, we insert this part:

2.4. Agar diffusion method

The antimicrobial activity of IP6 was evaluated by means of agar-well diffusion assays, as described by Valgas et al. [43]. Fifteen milliliters of the molten agar (45 °C) were poured into sterile petri dishes (Ø 90 mm). Working cell suspensions were prepared at 106 CFU/mL, and 100 μl was evenly spreaded onto the surface of the agar plates of Luria–Bertani (LB) agar (Oxoid Ltd, UK). Once the plates had been aseptically dried, 06 mm wells were punched into the agar with a sterile Pasteur pipette. IP6 was dissolved in water to a final concentration of 50 mg/ml and then filtered through 0.22 μm pore-size black polycarbonate filters (Millipore). Thus, 50 μl were placed into the wells and the plates were incubated at 37 °C for 24 h for bacterial strains. Antibacterial activity was evaluated by measuring the diameter of circular inhibition zones around the well. The un-inoculated media were also tested for inhibitory zones as a control. Tests were performed in triplicate.

Please see the revised version.

• L 152: Please adapt the output style to the requirements of the journal

Dear Reviewer, the correction was made as proposed. 

Please see the revised version.

• L 159: CFU per …?

Dear Reviewer, the correction was made as proposed. 

Please see the revised version.

• L 182: measuring the inhibition zones in AWDA?

Dear Reviewer, you are quite right, corrections are introduced. 

The antagonistic activity was assessed by determining the MIC values in comparison with CA and by measuring the inhibition zones by agar diffusion method. 

Please see the revised version.

• L 203: indicative strains- indicator strains?

Dear Reviewer, the correction was made as proposed. 

Please see the revised version.

• LL 259-262: To my opinion, the data do not prove chelat forming. Perhaps you could rebuilt the sentence as: According to Kimm and Rhee this effect might by due to the strong chelating …. ?

Dear Reviewer, the correction was made as proposed. 

Please see the revised version.

• LL 274 The ANCOVA and later on WALD Z statistic was not mentioned in the material and method section.

Dear Reviewer, correction was made according to your remark 

• The following paragraph was revised in the Materials and Methods part in the 2.6. Statistical analyses section. According to Reviewer comment we insert this part:

…. Mixed models were fitted using SPSS 19 and followed by post hoc contrasts through the origin. The interpretation of the statistical output by Analysis of Covariance (ANCOVA, SPSS; covariates, time and trial) of a mixed model requires an understanding of how to explain the relationships among the fixed and random effects in terms of the hierarchy levels. The significance or not of all estimates was confirmed by Wald Z. 

Please see the revised version.

• L285-286: I don’t think, that you have to explain the meaning and interpretation of P.

Dear Reviewer, Thank you for the pertinent remark. In fact, an upper decision limit is obtained using adjusted P-value Also, in our study; the test of homogeneity of variances gives P-value 0.05 that supports homogeneity of variances Equally, as indicated in the ‘Statistical analyses’ section in Materials and Methods part, the probability level of P<0.05 was used for assessing the statistical significance of the experimental data, and Tukey's post hoc test was used to determine whether differences between each of the mean values were significant (P<0.05). 

• LL 313-315 Well, yes - but this sentence is confusing.

Dear Reviewer, you are quite right, this sentence was removed from the text

Please see the revised version.

---

## [Editor Report · Decision Letter 1]

24 Mar 2020

Towards understanding the antagonistic activity of phytic acid against common foodborne bacterial pathogens using a general linear model

PONE-D-19-34291R1

Dear Dr. Hichem Chouayekh,

We are pleased to inform you that your manuscript has been judged scientifically suitable for publication and will be formally accepted for publication once it complies with all outstanding technical requirements.

With kind regards,

Filippo Giarratana

Academic Editor

PLOS ONE

---

## [Editor Report · Acceptance letter]

2 Apr 2020

PONE-D-19-34291R1 

Towards understanding the antagonistic activity of phytic acid against common foodborne bacterial pathogens using a general linear model 

Dear Dr. Chouayekh:

I am pleased to inform you that your manuscript has been deemed suitable for publication in PLOS ONE. Congratulations! Your manuscript is now with our production department. 

With kind regards,

on behalf of

Dr. Filippo Giarratana 

Academic Editor

PLOS ONE